# The Machinability of Flat-Pressed, Single-Layer Wood-Plastic Particleboards while Drilling—Experimental Study of the Impact of the Type of Plastic Used

Jarosław Górski *, Piotr Podziewski and Piotr Borysiuk

Institute of Wood Science and Furniture, Warsaw University of Life Science, Nowoursynowska 159, 02-787 Warsaw, Poland; piotr_podziewski@sggw.edu.pl (P.P.); piotr_borysiuk@sggw.edu.pl (P.B.)
* Correspondence: jaroslaw_gorski@sggw.edu.pl

**Abstract:** Machinability testing of ordinary wood-based panels can be useful, but testing prototypical (not produced industrially) panels is even more useful. So, the innovative (made only on a laboratory scale) flat-pressed WPCs were the subject of this study. The study consisted of experimental machinability testing of samples of fourteen different types of particleboards. Nine of them were innovative (non-commercial by design) particleboards, which differed from each other in terms of the type of plastic that was used and its percentage. The wood particles were bonded with either polyethylene (PE), polystyrene (PS) or polypropylene (PP). The percentages of plastic were either 30%, 50% or 70%. The research stand used for testing the machinability while drilling was based on a standard CNC (computerized numerical control) machining center. The experimental procedure involved the use of a specialized, accurate system for measuring cutting forces. Moreover, the maximum widths of the damage zones visible around the hole, on the drill entry side and the drill exit side were monitored using a digital camera and graphical software. Two key relative machinability indices were determined (quality problem index and cutting force problem index). Generally, the machinability of wood–polypropylene (W-PP) and wood–polystyrene (W-PS) composites was relatively good and generally similar both to each other and to the machinability of raw, standard particleboard P4. However, wood–polyethylene (W-PE) composite turned out to be the best wood-based board that was tested (even better than standard *MDF*) from the point of view of the cutting force criterion. On the other hand, the general quality of the holes made in W-PE composite was very poor (not much better than for raw, standard particleboard P5, but clearly better than for standard OSB).

**Keywords:** wood–plastic composite; drilling; relative machinability; hole quality





## 1. Introduction

There are many ways to machine particleboards [1,2] and therefore the machinability of these materials can be tested in many ways. Despite this fact, the machinability of any type of particleboards in the case of drilling is one of the most important issues from the practical point of view. This general belief can be proved in many ways, but it seems that it is enough to recall two basic arguments.

Firstly, the resistance to axial withdrawal of screws is one of the most essential, technical parameters of particleboards characteristics. This resistance should be determined experimentally in accordance with the detailed standard [3]. The experimental procedure requires the prior drilling of an appropriate hole in order to mount the screw, so drilling is a very basic form of particleboards machining.

Secondly, drilling is now used for more than just making holes for construction purposes. Drilling-based tests are commonly considered the most convenient (the quickest and the most material-saving) methods of relative machinability rating of any wood or wood-based materials [4–6].

Any scientific research on machinability has to be strictly experimental. In general, it has long been known that all attempts to theoretically determine (forecast) the machinability based on the mechanical properties of the material are useless [7]. This may contradict the belief that, for example, the knowledge of tool geometry, cutting parameters and standard properties of the material allows for theoretical determination of cutting forces. The research to date shows that real cutting processes, such as drilling, are too complicated from a physical point of view to find a direct relationship between the cutting forces and the tensile or shear strength of the material being machined [4,5]. We are simply forced to carry out experimental research.

Unfortunately, there is no generally accepted standard that could be directly applied to testing particleboard machinability. One of the most reliable testing procedures (which can be used for drilling in wood-based boards) has been suggested and tried by Podziewski et al. [4,5]. The procedure considers two basic aspects (criteria) of machinability: the hole quality and the cutting force. This is due to the fact that Podziewski et al. [4] (after consulting with scientists dealing with cutting theory and woodworking engineers) found that these are the only two fundamental criteria that matter in case of drilling in wood-based materials. The problem of the machining quality can significantly limit the scope of application of a construction material and the excessive drilling resistance may make it necessary to limit the feed speed and reduce machining efficiency. Using the procedure suggested by Podziewski et al. [4], the two key relative machinability indices of fourteen ordinary wood-based panels can be determined:

- Quality problem index (*QPI*);
- Cutting force problem index (*CFPI*).

Machinability testing of ordinary wood-based panels can be useful, but testing prototypical (not produced industrially) panels is even more useful. When an inventor designs a new type of particleboard (or an investor buys one that is unknown to him), he would like to know what its machinability is compared to some standard wood-based materials with much better-known characteristics [5]. It can be argued whether wood-plastic composites (WPCs) are really innovative materials as of now.

At the beginning, it is worth justifying the idea of mixing fossil plastics and renewable bio-materials. From the point of view of natural wood enthusiasts, the production of WPCs may seem like a step backwards in terms of design or ecology. However, globally, the exact opposite is the case for at least three important reasons.

WPCs with an equal share (50:50) of wood and thermoplastics are a step forward (compared to the standard plastics that are still used), because they reduce the use of plastics from hydrocarbon fossil resources by 50%. This contributes to the sustainable economic growth [8]. Besides, new design concepts can be developed.

The production of WPC can use post-consumer plastics and be a cost-effective method of recycling these plastics [9].

At the end of their service life, WPCs can be recycled and used again. "WPCs can be recycled several times" and the laboratory-tested "methodology can be industrially adapted for the manufacturing of recycled products" [8].

Therefore, WPCs are becoming more and more common and can be used in furniture (and a lot of other consumer goods) manufacturing or in building construction [10]. On the other hand, their properties are constantly intensively researched because they are still not as well-known as ordinary particleboards. Moreover, very different (and very innovative) WPCs can be manufactured on a laboratory scale. According to the current state of knowledge, WPCs are generally characterized by lower modulus of elasticity (MOE) and modulus of rupture (MOR), with comparable tensile and compressive strengths compared to standard wood materials [11–13]. An important advantage of WPCs compared to other wood-based panels is their resistance to moisture, which increases with the increase in the content of thermoplastics [11–13]. WPCs can also be processed using typical tools and woodworking machines. Zbieć et al. [14,15] found that cutting parameters used for ordinary (glued with UF resin) particleboards turned out to be also useful for WPCs



machining. A decent machinability of WPCs was also confirmed by Wilkowski et al. [16], and Borysiuk et al. [13]. Wilkowski et al. [16] also showed a decrease in cutting forces during drilling processing of WPCs in relation to three-layer particleboards. Among the analyzed composites, the greatest reduction in forces was obtained for composites bonded with polyethylene (PE), and additionally, a decrease in forces was demonstrated with an increasing share of thermoplastic. Borysiuk et al. [13] showed that cutting forces during the drilling or milling of WPCs were lower than the forces measured during the machining of ordinary particleboards. Somsakova et al. [17], when examining the quality of turning of the WPC, did not confirm the expected reduction in the machining quality with an increase in the feed rate. This was explained by the random distribution of the individual components of the composite on the cross-section of the workpiece.

The most common WPCs may differ in the type and percentage of plastic (although the most common ratio of wood to plastic in commercial WPCs is about 50/50), but they are usually produced by extrusion or injection methods, that is, by means of technological lines analogous to ones that are used in the production of plastic products. However, the alternative idea of WPC manufacturing, based on the concept of bonding wood chips by using thermoplastics is also being developed. So called flat-pressed WPCs can be successfully manufactured using methods such as the traditional technology of ordinary particleboards [18–22]. Such type of innovative boards (made only on a laboratory scale at the Warsaw University of Life Sciences) were the subject of this study. Their machinability in case of drilling was tested for the first time using the full procedure, which has been suggested and tried by Podziewski et al. [4,5]. This is the main novelty of this study. Admittedly, attempts were made to test the machinability of flat-pressed WPCs, but never in such a systematic and comprehensive way. Until now, researchers have generally limited themselves to measuring cutting forces, completely ignoring the problem of cutting quality. It may not be enough because both of these aspects should be investigated simultaneously, because only this gives a complete picture of the issue. In addition, the machinability of flat-pressed WPCs has never been compared with that of a wider range of standard particleboards. The objective of this study was to fill the aforementioned knowledge gap.

The main objectives of the study were as follows:

- Experimental determination of the relative machinability indices (quality problem index and cutting force problem index) for WPCs and a few standard particleboards in order to compare them with each other in this regard;
- Checking whether the type and percentage of plastic used in WPCs have a statistically significant effect on the machinability.

## 2. Materials and Methods

The study consisted of experimental machinability testing of samples of fourteen different types of particleboards. Five of them were strictly commercial products: raw medium-density fiberboards (*MDF*), raw three-layer particleboards P4 and P5 (classification according to [23]), melamine faced particleboard P3 (also according to [23]), and raw oriented strand board (OSB). All samples came from wood-based boards produced on a mass scale and targeted at the European Union market. More detailed characteristics of these materials are shown in Table 1. Moreover, the machinability of samples of nine other (non-commercial by design) particleboards was tested. These prototype (experimental) particleboards were made, only in small amounts, in the laboratory of the Institute of Wood Sciences and Furniture (Warsaw University of Life Sciences). They were single-layer, flat-pressed wood-plastic composites. They were produced in nine different variants, which differed from each other in terms of the type of plastic that was used and its percentage. The wood particles were bonded with polyethylene (PE), polystyrene (PS) or polypropylene (PP). The percentages of plastic were: 30%, 50% or 70%. More detailed characteristics of these materials are shown in Table 2.

**Table 1.** Detailed Information about the Tested Wood-Based Boards.

| Board Type | Density (kg/m³) (Standard Deviation) | Modulus of Rupture (MOR) (N/mm²) (Standard Deviation) | Modulus of Elasticity (MOE) (N/mm²) (Standard Deviation) | Brinell Hardness (HB) (Standard Deviation) | Main Application |
|---|---|---|---|---|---|
| Raw *MDF* | 746 (7.75) | 33.9 (2.45) | 4180 (106) | 4 (0.07) | Furniture components: frames, doors |
| Raw particleboard P4 | 649 (4.39) | 13.1 (0.71) | 3204.4 (75) | 2.6 (0.20) | Furniture components: upholstered furniture frames |
| Melamine faced particleboard P3 | 666 (6.46) | 15.4 (1.66) | 2948.4 (37) | 2.1 (0.05) | Furniture components: frames, doors |
| Raw particleboard P5 | 725 (17.51) | 21.1 (1.09) | 3802.9 (108) | 4.7 (0.04) | Furniture industry, construction |
| OSB | 595 (25.98) | 30.9 (3.57) | 5490.1 (133) | 4.2 (0.41) | Building construction, flooring |

**Table 2.** Detailed Information about the Tested Wood-Based Boards.

| Feature | | | | Particleboard Type | | | | | |
|---|---|---|---|---|---|---|---|---|---|
| | Polyethylene Bonded (PE) | | | Polystyrene Bonded (PS) | | | Polypropylene Bonded (PP) | | |
| **Thermoplastic Ratio** | **30%** | **50%** | **70%** | **30%** | **50%** | **70%** | **30%** | **50%** | **70%** |
| Density (kg/m³) (Standard deviation) | 650 (20) | 672 (24) | 670 (22) | 652 (24) | 675 (33) | 668 (28) | 655 (22) | 669 (24) | 652 (28) |
| MOR (N/mm²) (Standard deviation) | 8.6 (0.7) | 11.6 (0.8) | 11.3 (0.7) | 9.9 (0.8) | 18.6 (1.4) | 20.4 (1.1) | 13.3 (1.4) | 17.9 (1.8) | 16.5 (4.1) |
| MOE (N/mm²) (Standard deviation) | 1006 (18) | 1141 (63) | 870 (71) | 1330 (145) | 2022 (70) | 1681 (129) | 1619 (24) | 1651 (169) | 1364 (227) |
| Internal bond (N/mm²) (Standard deviation) | 0.62 (0.08) | 0.93 (0.05) | 1.49 (0.11) | 0.52 (0.07) | 0.86 (0.09) | 1.68 (0.12) | 1.73 (0.12) | 1.29 (0.10) | 1.25 (0.10) |
| Screw holding (N/mm²) (Standard deviation) | 84 (9.0) | 90.5 (10.2) | 83.5 (9.4) | 108 (11.1) | 135 (12.3) | 167 (14.2) | 113 (10.3) | 153 (13.1) | 154 (13.5) |
| Thickness swelling after 24 h (%) (Standard deviation) | 23.2 (3.4) | 9.6 (1.0) | 3.8 (0.5) | 31 (3.7) | 5.4 (0.8) | 2.2 (0.3) | 19.5 (2.6) | 6 (0.7) | 1.5 (0.2) |
| Water absorption after 24 h (%) (Standard deviation) | 80 (9.0) | 52 (6.3) | 34.7 (4.8) | 88.9 (9.3) | 56.2 (5.7) | 34.4 (4.2) | 57.3 (7.1) | 30.4 (4.0) | 19.5 (2.8) |

The data shown in Tables 1 and 2 were determined experimentally according to appropriate standards. The average density was determined in accordance with [24], the bending modulus and bending strength in accordance with [25], hardness in accordance with [26], screw pull resistance in accordance with [3], swell and absorbability after 24 h soaking in water in accordance with [27]. The general information about the basic mechanical properties of the thermoplastics used (quoted from specialized literature [28–30] is show in Table 3.

**Table 3.** General information about the basic mechanical properties of the thermoplastics used [28–30].

| Feature | Thermoplastic Type | | |
| --- | --- | --- | --- |
| | Polyethylene (PE) | Polystyrene (PS) | Polypropylene (PP) |
| Density (kg/m$^3$) | 915–935 | 1040–1060 | 900–920 |
| MOR (N/mm$^2$) | 8–23 | 40–70 | 21–37 |
| MOE (N/mm$^2$) | 200–500 | 3100–3300 | 1100–1300 |

The research stand used for machinability testing in case of drilling was based on a standard CNC (computerized numerical control) machining center (Busellato Jet 100). A typical one-blade drill bit intended for through drilling in wood-based panels was used. The tool diameter was 10 mm, and the blade was made out polycrystalline diamond. The drill was a commercial product (Leitz ID 091193). The general view of the drill bit with a tool holder is shown in Figure 1. All of the holes in all of the boards were drilled with one spindle speed (6000 rpm) and with one feed per revolution (0.15 mm/rev). For each of the tested materials, a series of 20 holes were performed. The experimental results were used to determine the relative machinability indexes based on the following criteria: machining quality and on the cutting forces as criteria. Machinability testing procedure and definitions of relative machinability indexes were generally in line with the procedure suggested and tried by Podziewski et al. [1]. The only slight difference was that the research was limited to one (most recommended in practice) aforementioned feed per revolution value (0.15 mm/rev).

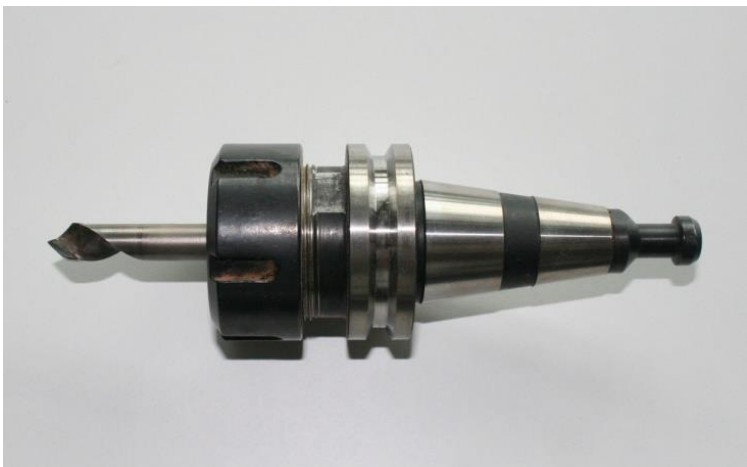

**Figure 1.** General view of the drill bit (shank part diameter—10 mm; shank length—27 mm; total length—70 mm; number of teeth—1) with a tool holder (ISO 30, ER 32).

One of the primary objectives of the study was to monitor and compare the hole quality (defined as a quality of the edges of the holes), which was seen on both sides of the drilled boards (i.e., on the side where the drill bit entered the materials and where it came out). A digital camera (Canon 40D) with an appropriate lens (Canon Macro Lens EF 100 mm 1: 2.8 USM) was used to monitor the hole quality. Sample photos of the hole taken on both sides of the particleboard are shown in Figure 2.

The photographs were analyzed using CorelDRAW Graphics Suite graphical software. This way, the maximum width of the damage zone (visible around the hole on both sides of the board) was measured for each hole. When looking for the border of the damage zone, the researchers analyzed any signs of surface destruction such as fiber pullout, fiber fragmentation, burr, bulge or structure crack. After locating the border of the external damage area, the characteristic diameter (marked with the symbol $D$ and shown in Figure 3) of the specific circle was determined. The center of this specific circle was on the symmetry axis of the hole and the circle passed through the farthest boundary point (Figure 3). The

maximum width of the damage zone (marked with the symbol *W*) was calculated using the following formula:

$$W = (D - N)/2 \tag{1}$$

where *N* stands for the nominal diameter of the drill bit (*N* = 10 mm).

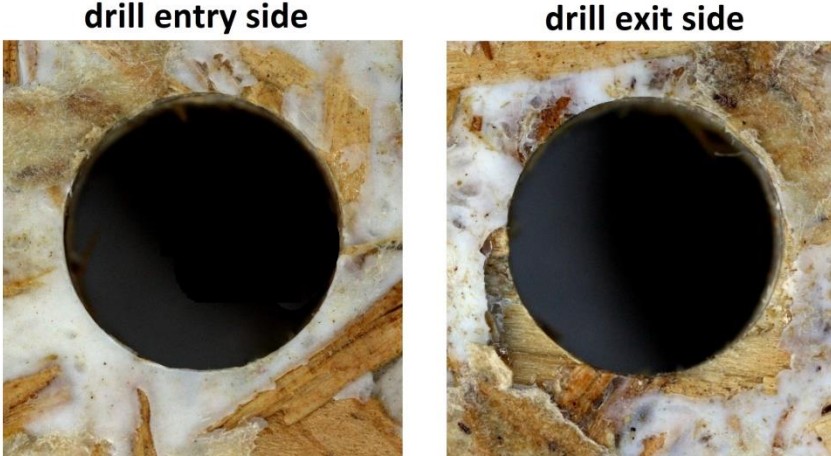

**Figure 2.** Sample photos of the hole taken on both sides of the particleboard.

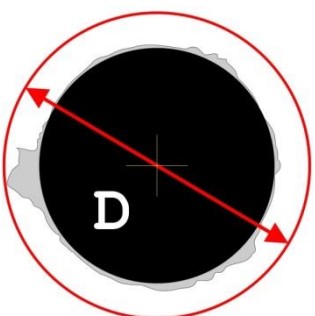

**Figure 3.** The characteristic diameters *D* determined on both sides of the boards during the machining quality testing.

To be more precise, two different characteristic diameters were determined for each hole. One of them was determined on the side where the drill bit entered the materials and was marked with the symbol *D*1. Characteristic diameter determined on the opposite side of the board was marked with the symbol *D*2. Analogously, two different widths of the damage zone were calculated for each hole (they were marked with symbols *W*1 and *W*2, respectively).

The width values marked as *W*1 and *W*2 were averaged separately for all holes made in the same material.

Next, a relative index called the quality problem index (*QPI*) was calculated for each material according to following formula:

$$QPI_X = 0.5 \times ((W1_X/W1_{MDF}) + (W2_X/W2_{MDF})) \times 100\% \tag{2}$$

where $QPI_X$ index defines the relative difficulty in machining of *X* material due to problems with the quality of the machining (the lower the index the better machinability of the *X* material), $W1_X$ and $W2_X$ describe the width of the damaged zone concerning *X* material, and $W1_{MDF}$ and $W2_{MDF}$ denote the analogues width for the reference material for which the raw *MDF* was taken.

Another objective of the study was to measure and compare the feed force and torque, generated when drilling holes in various materials. Therefore, the experimental procedure

involved the use of a specialized, accurate system for measuring cutting forces. The scheme of the system is shown in Figure 4. The force-torque measuring system was built using a special platform based on the piezoelectric dynamometer using a 2-component sensor (Kistler 9345, Winterthur, Switzerland). This sensor was designed to monitor the feed force (F) and the cutting torque (T). The measurement system also included other elements: a signal amplifier (Kistler ICAM 5073A, Winterthur, Switzerland), a connector block (NI BNC-2110, Austin, TX, USA), and a data acquisition system (NI PCI-6034E, Austin, TX, USA). All recorded signals were analyzed in NI LabVIEW environment. Based on the recorded feed force and torque signals, their average values were determined during "the main drilling phase". The additional figure (Figure 5) shows typical changes in the feed force signal to explain what the "the main drilling phase" was.

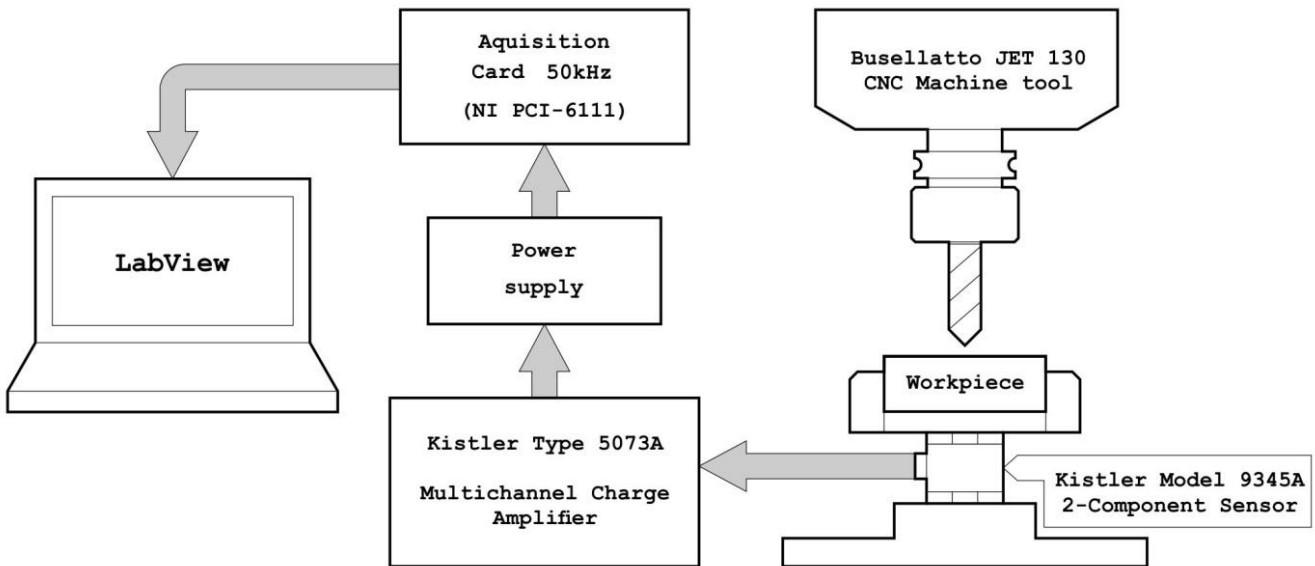

**Figure 4.** The scheme of the system used for measuring cutting forces during the machining testing.

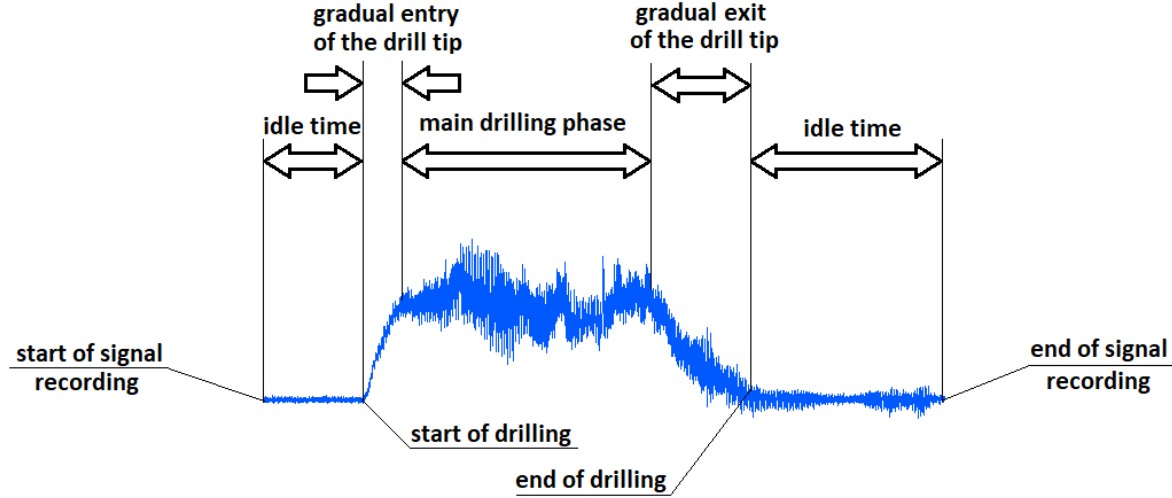

**Figure 5.** Typical changes in the feed force signal recorded during the experiments.

A relative index called the cutting force problem index (*CFPI*) was calculated for each material according to following formula:

$$CFPI_X = 0.5(F_X/F_{MDF} + T_X/T_{MDF}) \times 100\% \qquad (3)$$

where $CFPI_X$ index defines the relative difficulty in machining of $X$ material regarding the size of the cutting forces (the lower the index, the better machinability of the $X$ material), $F_X$ and $T_X$ denote the feeding force and torque for the $X$ material, and $F_{MDF}$ and $T_{MDF}$ describe the feeding force and torque of the reference material for which the raw *MDF* was taken.

After the experiment was completed, the standard analysis of variance (ANOVA) was used to analyze the effect of the plastic type and the wood-plastic ratio on the machinability of the flat-pressed WPCs. ANOVA was used to determine whether there are any statistically significant differences between sample groups that need to be compared. In cases where it was found that not all of the group means are equal, the post hoc tests (based on Tukey method) were run to identify which particular differences between pairs of means are really significant.

## 3. Results and Discussion

The results of experimental studies on the effect of the plastic type and the wood-plastic ratio on the quality of the holes (which were analyzed on both sides of the drilled-right-through boards), are shown in Figure 6. The results of using two-way ANOVA to verify the statistical significance of the impact of these factors are shown in Tables 4 and 5. It turned out that both of these factors had a statistically significant impact on the hole quality, which was monitored on the side of the drill entry into the material (*W*1). From this point of view, everything seems to be clear—polystyrene (PS) was the most advantageous plastic. However, the results of the standard post hoc analysis (Tukey test) showed that there was no statistically significant difference between wood–polystyrene and wood–polypropylene composites, but both of these materials had a significant advantage over wood–polyethylene. Likewise, it seems that the more plastic, the better the quality; however, there was no statistically significant difference between the 50% and 70% plastic content.

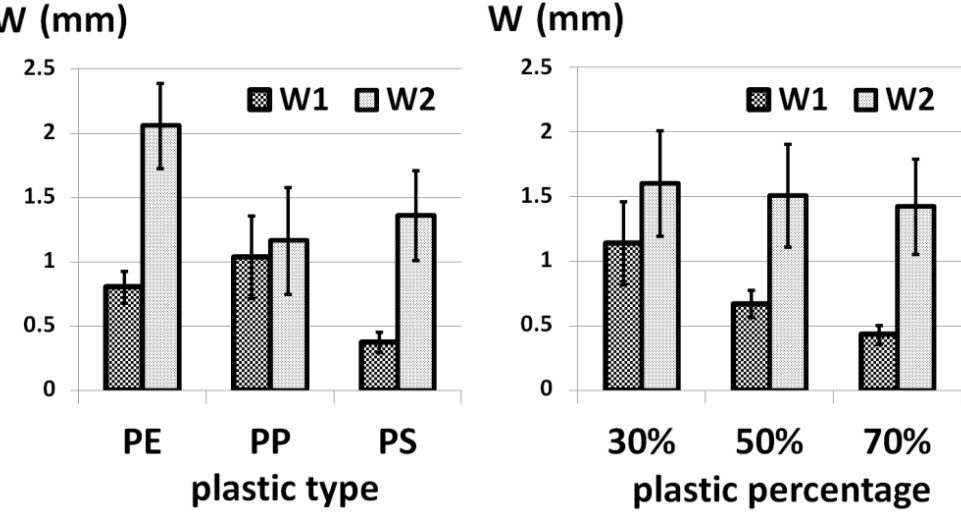

**Figure 6.** The effect of the plastic type and the wood–plastic ratio on the quality of the holes, which were analyzed on both sides of the boards (*W*1 [mm] and *W*2 [mm] are the maximum widths of the damage zones visible around the hole on the drill entry side and the drill exit side, respectively).

There are slightly different conclusions regarding the holes quality on the other side of the boards (*W*2). This time, polypropylene (PP) seemed a bit more favorable than polystyrene (PS), but the results of the post hoc test showed that there was no statistically significant difference between them. At the same time, the quality of holes made in the boards containing polyethylene (PE) looked extremely poor (compared to the other two plastics). The impact of the plastic percentage seemed to have a similar character on both sides of the boards (the more plastic, the better the quality), but it turned out that on the

drill exit side this tendency was not statistically significant. In any case, the hole quality on the drill exit side was significantly worse than on the drill entry side (as usual in drilling).

**Table 4.** The results of using ANOVA to verify the statistical significance of the impact of plastic type and plastic percentage on the maximum width of the damage zone on the drill entry side ($W1$ (mm)).

| Source | Sum Sq. | df | Mean Sq. | F | Prob > F |
|---|---|---|---|---|---|
| Plastic type | 6.7493 | 2 | 3.37465 | 14.89 | $2.84912 \times 10^{-6}$ |
| Plastic percentage | 7.5142 | 2 | 3.75709 | 16.58 | $8.31225 \times 10^{-7}$ |
| Error | 19.2591 | 85 | 0.22658 | | |
| Total | 33.5226 | 89 | | | |

**Table 5.** The results of using ANOVA to verify the statistical significance of the impact of plastic type and plastic percentage on the maximum width of the damage zone on the drill exit side ($W2$ (mm)).

| Source | Sum Sq. | df | Mean Sq. | F | Prob > F |
|---|---|---|---|---|---|
| Plastic type | 13.226 | 2 | 6.61308 | 6.45 | 0.0025 |
| Plastic percentage | 0.796 | 2 | 0.39813 | 0.39 | 0.6795 |
| Error | 87.192 | 85 | 1.02578 | | |
| Total | 101.214 | 89 | | | |

The results of experimental studies on the effect of the plastic type and the wood–plastic ratio on the drilling torque and feed force, are shown in Figures 7 and 8, respectively. The results of using two-way ANOVA to verify the statistical significance of the impact of these factors are shown in Tables 6 and 7. It turned out that both factors were significant in relation to torque but only one of them (plastic type) influenced the feed force. The results of the Tukey test clearly showed that the smallest cutting forces arose during drilling holes in the boards containing polyethylene (PE). Wood–plastic composites made using polypropylene (PP) and polystyrene (PS) seemed to differ from each other in this respect (polystyrene seemed a bit more favorable), but the post hoc analysis showed that it was not a statistically significant tendency.

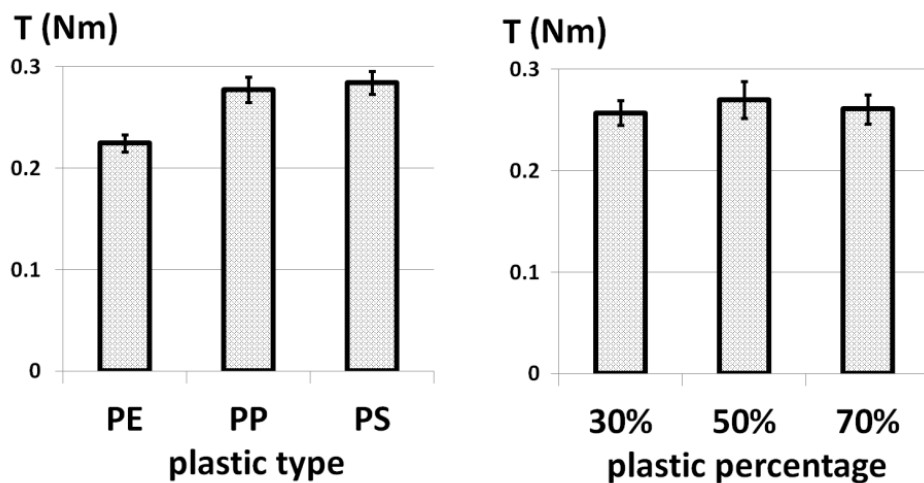

**Figure 7.** The effect of the type of plastic and the wood–plastic ratio on the drilling torque (T [Nm]).

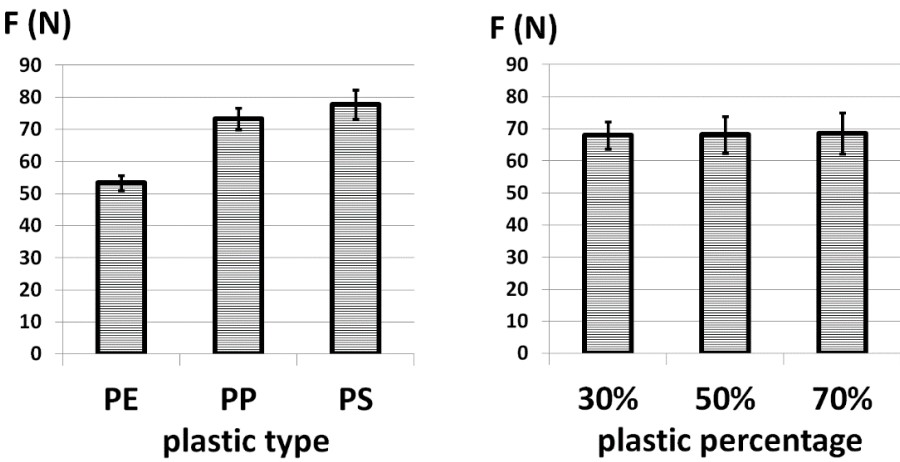

**Figure 8.** The effect of the type of plastic and the wood–plastic ratio on the feed force (F [N]).

**Table 6.** The results of using ANOVA to verify the statistical significance of the impact of plastic type and plastic percentage on the on the drilling torque (T (Nm)).

| Source | Sum Sq. | df | Mean Sq. | F | Prob > F |
|---|---|---|---|---|---|
| Plastic type | 64,173.7 | 2 | 32,086.8 | 86.33 | 0 |
| Plastic percentage | 5192.9 | 2 | 2596.5 | 6.99 | 0.0012 |
| Error | 64,669.4 | 174 | 371.7 | | |
| Total | 134,424.7 | 178 | | | |

**Table 7.** The results of using ANOVA to verify the statistical significance of the impact of plastic type and plastic percentage on the on the feed force (F (N)).

| Source | Sum Sq. | df | Mean Sq. | F | Prob > F |
|---|---|---|---|---|---|
| Plastic type | 18,988,422 | 2 | 949,421.2 | 120.89 | 0 |
| Plastic percentage | 2992.18 | 2 | 1496.1 0 | 2.19 | 0.8267 |
| Error | 13,664,941 | 174 | 7853.4 | | |
| Total | 3,268,966 | 178 | | | |

The general differentiation of the five standard wood-based boards and the three innovative wood–plastic particleboards in terms of difficulty in drilling, determined by two independent indexes (*QPI* and *CFPI*, which have been defined by mathematical Equations (2) and (3) in Section 2, is shown in Figure 9. This figure shows the machinability of wood–polyethylene (W-PE), wood–polypropylene (W-PP) and wood–polystyrene (W-PS) composites with 50% plastic content. Such a ratio of wood to plastic was considered most representative because it is the most common ratio in commercial WPCs. Referring to Figure 9 (which is a two-dimensional machinability chart), the machinability of the innovative particleboard and the standard particleboards (e.g., P3, P4, P5, and OSB) or standard medium-density fiberboard (*MDF*) can be compared. The quality criterion (*QPI*) differentiated the tested materials considerably more than the cutting force criterion (*CFPI*). The *QPI* varied from 100% to almost 500%, and the *CFPI* varied from 70% to 130%. It turned out that the machinability of W-PP and W-PS composites was relatively good and generally alike each other and alike the machinability of raw, standard particleboard P4. The machinability indicators of W-PE were completely different. W-PE turned out to be the best out of the wood-based boards (even better than standard *MDF*) from the cutting force criterion standpoint. On the other hand, the quality of the holes made in this material was very poor (not much better than in raw, standard particleboard P5).

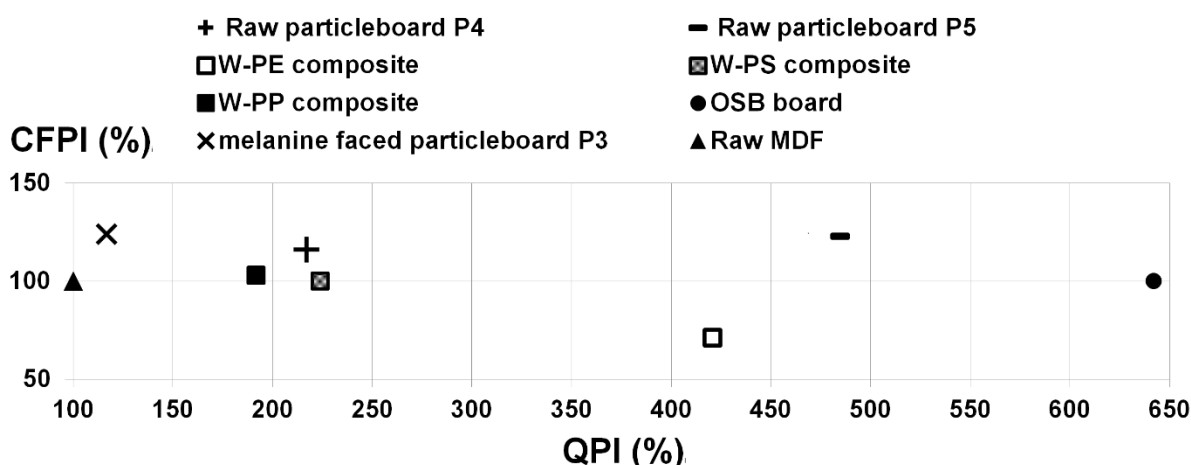

**Figure 9.** Differentiation of five standard wood-based boards and three innovative wood–plastic particleboard in terms of difficulty in drilling, determined by two independent indexes (*QPI* and *CFPI*, which have been defined by mathematical Equations (2) and (3) in Section 2).

These conclusions regarding the differences in machinability between W-PP, W-PS and W-PE according to the cutting force criterion are generally consistent with the results of previous studies, e.g., [10,13] (although it was not previously suggested that W-PE might be so clearly superior to standard *MDF* in this regard). A more direct comparison of the results of the current and previous studies is not possible due to different methodological assumptions. Moreover, the results of the research on machinability according to the hole quality criterion cannot be confronted with a different point of view because there are no analogous scientific publications on this subject as of now.

The results of the experimental studies discussed so far should be subject to a slightly more in-depth analysis. First of all, it is worth wondering why the machinability of W-PE composite differs so much from the other two materials (W-PP and W-PS composites). At the beginning, the information about the basic mechanical properties of these WPCs and thermoplastics (Tables 2 and 3) should be recalled. As was already mentioned in the introduction, there is no direct relationship between the cutting forces and the standard mechanical properties of the material being machined. However, on the other hand, some indirect relationships can be expected. The data in Table 2 shows that W-PE composite clearly has worse mechanical properties (MOR and MOE) than W-PP or WP-PS composites. It is probably connected with the fact that ordinary PE has worse mechanical properties (MOR and MOE) than ordinary PP or PS (Table 3). In this situation, the fact that the cutting forces generated during drilling in W-PE composite were significantly lower than for W-PP and W-PS composites seems quite understandable. It is far riskier to try to explain why drilling in W-PE composite creates a greater width of the damage zone (visible around the hole) compared to drilling in W-PP or W-PS composites. Perhaps this is because stiffer material (material with higher values of MOE and MOR) is easier to cut without irregular deformation and rupturing of the particles, but this is just a hypothesis.

Of course, the research presented in this paper has significant limitations. Firstly, it concerns only three thermoplastics. We would like to include other thermoplastics and also some thermoset plastics in further research. It is definitely worth looking for new plastics that could form a strong bond with wood particles due to good adhesion. In addition, it would certainly be worth experimenting with optimizing the manufacturing process of flat-pressed WPCs.

## 4. Conclusions

- The results of the study suggest that both the type of plastic and the percentage of plastic were significant factors in relation to hole quality on the drill entry side, but only one of them (the type of the plastic) influenced the quality on the drill exit side.

From this point of view, polystyrene (PS) and polypropylene (PP) were far more advantageous plastics than polyethylene (PE). In general, the more plastic, the better the quality on the drill entry side, but there was no statistically significant difference between the 50% and 70% plastic content.

- Analogously, both the plastic type and the plastic percentage were significant factors in relation to torque, but only one of them (the plastic type) influenced the feed force. From this point of view, polyethylene (PE) was a far more advantageous plastic than polystyrene (PS) or polypropylene (PP). The last two plastics (PS and PP) did not differ significantly from each other.
- Generally, W-PP and W-PS composites were alike, and the machinability of W-PP and W-PS composites was relatively good, similar to the machinability of raw, standard particleboard P4. However, W-PE composite turned out to be the best wood-based board out of all the tested ones (even better than standard *MDF*) from the point of view of the cutting force criterion. On the other hand, the general quality of the holes made in W-PE composite was very poor (not much better than for raw, standard particleboard P5, but clearly better than for standard OSB).
- It is worth considering other plastics (not only other thermoplastics, but also thermoset polymers) in further research.

**Author Contributions:** Conceptualization, J.G. and P.B.; methodology, J.G. and P.P.; validation, J.G. and P.B.; formal analysis, J.G. and P.P.; investigation, P.P. and P.B.; re-sources, P.B.; data curation, P.P. and J.G.; writing—original draft preparation, J.G. and P.B.; writing—review and editing, J.G. and P.B.; visualization, P.P.; supervision, P.B. and J.G.; project administration, P.P. and J.G.; funding acquisition, P.B. and P.P. All authors have read and agreed to the published version of the manuscript.

**Funding:** This research received no external funding.

**Institutional Review Board Statement:** Not applicable.

**Informed Consent Statement:** Not applicable.

**Data Availability Statement:** The data presented in this study are available on request from the authors.

**Conflicts of Interest:** The authors declare no conflict of interest.

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
