# Peer review of "The Machinability of Flat-Pressed, Single-Layer Wood-Plastic Particleboards while Drilling—Experimental Study of the Impact of the Type of Plastic Used"

_forests, doi:10.3390/f13040584_

Round 1

Reviewer 1 Report

Dear Authors,

I have reviewed your article an titled "The Machinability of flat-Pressed, Single-Layer Wood-Plastic Particleboards while Drilling – Experimental Study of the Impact of the Type of Plastic Used". The article discusses a very popular machining process used in woodworking and machining wood-based materials, holes drilling.

An important issue is the quality of holes in drilling process for wood-based materials. In this article proposed interesting parameters to rate of machinability properties. However, the article needs a several corrections and additions, as follow:

# introduction is good, but it relies almost entirely on authors self-citations. Perhaps this chapter should cover the width discussed machinability properties particleboards. Below two examples of references:

PaÅ‚ubicki, B.; Hlásková, L.; RogoziÅ„ski, T. Influence of Exhaust System Setup on Working Zone Pollution by Dust during Sawing of Particleboards. Int. J. Environ. Res. Public Health 2020, 17, 3626. https://doi.org/10.3390/ijerph17103626

Pałubicki, B. Cutting Forces in Peripheral Up-Milling of Particleboard. Materials 2021, 14, 2208. https://doi.org/10.3390/ma14092208

# materials and methods

Tables 1 and 2 present the mechanical and structural properties tested materials. These properties were determined according to standards. Probably in tables are shown average values, but there also should be shown sprits of values or standard deviations.

Standards presented in the text should also be added to the reference list

A general view at the machine tool is not necessary. Enough will be information about model and manufacture company.

Figure 2. there should be marked cutting tool and tool holder. There also should be added information about type of tool holder taper (probably SK 40 according iso 69871) and kind of holding cutting tool in tool holder (probably a collet ER 20). This information is very important, because these parameters have effect on quality of main movement of cutting tool. In spite of the model and manufacturer of the drill bit is given, its basic geometrical parameters have to be also given.

Figure 3. Sample view of the actual hole from both sides of the plate (cutting tool entry and exit) is missing.

#results

The results discussed should include sample records of cutting forces and torques from the tests carried out, with particular reference to and discussion of the cutting tool input and output sections. What force value was considered to determine the index, maximum, average, etc.?

Figures 5 and 7 replace commas with dots in the description of the numbers.

The list of references items is double numbered.

Reviewer 2 Report

This manuscript needs substantial changes before being considered for publication.

(1) The abstract lacks an informative explanation regarding materials and methods. It sounds like the authors jumped from the introduction to the results. Moreover, the abstract does not end up with a closing statement that leaves readers with a take-home message.

(2) The knowledge gap is not well-explained in the introduction.

(3) The objectives and hypothesis of this study must be clearly defined. Besides, the authors should clarify how the outcome of this research partially bridges the knowledge gap?

(4) Figures 6 and 9 are ANOVA tables that were copied and pasted from software. They should be represented as tables, not figures.

(5) Did authors carry out additional statistical tests (Duncan, Tukey, Dunnett following the two-way ANOVA test? If not, please bring reasons to justify it. If yes, please elaborate on the results of the statistical test.

(6) A comparison should be made between the results of this research and the findings of previous relevant studies.

(7) Research limitations must be elaborated in the discussion part and briefly mentioned in the conclusion part.

(8) Directions for future research must be discussed extensively in the discussion part and briefly in the conclusion part.

(9) Above all, the novelty of this research has not been mentioned. 

Overall, the manuscript in the current format requires a major revision.

Reviewer 3 Report

Comment Number 1:

Corresponding to unyielding, high crystallinity HDPE grades, at break, the castoff PE matrix has relatively high stiffness and low elongation by increasing the MC content, the extension further diminishes paired with a robust boost in elastic modulus. These discoveries are anticipated as both cellulose and aluminum act as rigid grouts rising the elastic modulus of the composites.

Comment Number 2:

At tall, MC load the gristly nature of the caulking governs the structure of the material and illustrations at 80 and 90 wt% MC can be more effectually compared to fibrous systems with polymeric binders or even to cellulosic sheets than to polymer-matrix composites.

Comment Number 3:

Thermoplastic polymers and polyolefins in precise are the most explored matrices for such composites, nonetheless, various polymers still can be suggested, including thermosets. The ability of the selected polymer to form strong, due to the highly polar and hydrophilic nature of cellulosic materials, well adhered interfaces with such fillers is a crucial element to define processing and additivities strategies in further research. It’s a good research direction for future research.

Round 2

Reviewer 1 Report

Dear Authors, thank you for take into account my suggestions to corrections of article. 

In my opinion paper in currently shape can be published.

Reviewer 2 Report

The revised manuscript is satisfactory and the authors addressed the comments. The manuscript can be now accepted for publication.